# Enhancing Gastric Cancer Therapeutic Efficacy through Synergistic Cotreatment of Linderae Radix and Hyperthermia in AGS Cells

**DOI:** 10.3390/biomedicines11102710

**Published:** 2023-10-05

**Authors:** Chae-Ryeong Ahn, Seung-Ho Baek

**Affiliations:** 1Department of Science in Korean Medicine, Graduate School, Kyung Hee University, Seoul 02447, Republic of Korea; cofud2917@naver.com; 2College of Korean Medicine, Dongguk University, 32 Dongguk-ro, Goyang-si 10326, Republic of Korea

**Keywords:** hyperthermia, gastric cancer, Linderae Radix, heat shock proteins, reactive oxygen species, combination therapy

## Abstract

Gastric cancer remains a global health threat, particularly in Asian countries. Current treatment methods include surgery, chemotherapy, and radiation therapy. However, they all have limitations, such as adverse side effects, tumor resistance, and patient tolerance. Hyperthermia therapy uses heat to selectively target and destroy cancer cells, but it has limited efficacy when used alone. Linderae Radix (LR), a natural compound with thermogenic effects, has the potential to enhance the therapeutic efficacy of hyperthermia treatment. In this study, we investigated the synergistic anticancer effects of cotreatment with LR and 43 °C hyperthermia in AGS gastric cancer cells. The cotreatment inhibited AGS cell proliferation, induced apoptosis, caused cell cycle arrest, suppressed heat-induced heat shock responses, increased reactive oxygen species (ROS) generation, and promoted mitogen-activated protein kinase phosphorylation. N-acetylcysteine pretreatment abolished the apoptotic effect of LR and hyperthermia cotreatment, indicating the crucial role of ROS in mediating the observed anticancer effects. These findings highlight the potential of LR as an adjuvant to hyperthermia therapy for gastric cancer. Further research is needed to validate these findings in vivo, explore the underlying molecular pathways, and optimize treatment protocols for the development of novel and effective therapeutic strategies for patients with gastric cancer.

## 1. Introduction

Gastric cancer poses a formidable global health challenge, especially in Asian countries, where its incidence and mortality rates are strikingly high [1,2,3]. Traditional treatment methods, such as surgery, chemotherapy, and radiation therapy, face obstacles, including adverse side effects, tumor resistance, and patient tolerance [4,5]. Consequently, innovative and effective therapeutic approaches need to be developed to overcome these obstacles to achieve positive outcomes in patients.

Cancer hyperthermia therapy, a technique that utilizes heat to selectively target and eliminate cancer cells, has emerged as a potential solution to overcome the limitations of current approaches of gastric cancer treatment [6,7,8,9]. However, its standalone application exhibits limited efficacy because it triggers a heat shock response, leading to the activation of heat shock factors (HSFs) and heat shock proteins (HSPs). Such heat shock-induced responses may lead to cell survival and resistance to therapy [10,11,12].

Linderae Radix (LR), a natural herb steeped in traditional medicinal history, is renowned for its thermogenic properties and has been harnessed to treat various gastrointestinal ailments, such as indigestion, abdominal discomfort, and bloating [13,14,15]. Recognizing the therapeutic potential of LR, we hypothesized that its integration with cancer hyperthermia therapy might counteract the shortcomings of hyperthermia treatment and amplify its therapeutic effectiveness.

Based on this hypothesis, our study was designed to investigate the synergistic effects of LR and hyperthermia treatment. We aimed to explore the possibility of boosting the anticancer capabilities of both treatment modalities while minimizing the activation of heat shock responses that could promote cell survival and tumor resistance. Furthermore, we sought to elucidate the molecular mechanisms underlying these observed effects to understand the potential benefits of this treatment.

## 2. Materials and Methods

### 2.1. Preparation of LR Extract

*Lindera strichnifolia* (Kwangmyeongdang Medicinal Herbs Co., Ltd. Ulsan, Republic of Korea) was ground and homogenized using a homogenizer. After soaking for 24 h at room temperature in 70% EtOH, the extract was produced. The resulting extract was filtered (pore size: 5 μm), concentrated under reduced pressure, and lyophilized to obtain a sample. Samples at concentrations of 80, 100, and 120 μg/mL were prepared using dimethyl sulfoxide (DMSO) (Samcheon Chemical, Seoul, Republic of Korea). The solutions were stored at 4 °C until further use.

### 2.2. Liquid Chromatography (LC)–Mass Spectrometry (MS) Analysis

Chromatographic analysis was performed using ultraperformance liquid chromatography-electrospray ionization/quadrupole time-of-flight high-definition mass spectrometry/mass spectrometry (UPLC-ESI-QTOF-MS/MS) to identify the chemical components of the ethanolic extract. The extract was shaken in 50% methanol using a vortex mixer for 30 s and sonicated for 10 min. The supernatants were filtered through a 0.2 μm hydrophilic polytetrafluoroethylene syringe filter (Thermo Scientific, Waltham, MA, USA). The filtrate was diluted to 20 mg/mL and transferred to an LC vial prior to use. The LC–MS system consisted of a Thermo Scientific Vanquish UHPLC system (Thermo Fisher Scientific, Sunnyvale, CA, USA) with an ACQUTY UPLC HSS T3 column (2.1 mm × 100 mm, 1.8 μm; waters) and a Triple TOF5600+ mass spectrometer system (QTOF MS/MS, SCIEX, Foster City, CA, USA).

The QTOF MS system was equipped with an electrospray ionization (ESI) source in positive and negative ion modes and used to complete the high-resolution experiment. The elution program for UHPLC separation, which employed 0.1% formic acid in water as eluent A and 0.1% formic acid in acetonitrile as eluent B, was as follows: 0–1 min, 5% B; 1–4 min, 5–15% B; 4–1 1 min, 15–35% B; 11–17 min, 35–50% B; 17–21 min, 50% B; 21–25 min, 50–100% B; and 25–29 min, 100% B and equilibration with 5% B for 4 min at a flow rate of 0.4 mL/min. The column temperature was 40 °C, and the autosampler was maintained at 4 °C. The injection volume of each sample solution was 2 μL. Data acquisition and processing for qualitative analysis were conducted using Analyst TF 1.7, PeakVeiw2.2, and MasterView (SCIEX, Foster City, CA, USA). The MS/MS data for qualitative analysis were processed using PeakView and MasterView to screen for probable metabolites based on accurate mass and isotope distributions.

### 2.3. Cell Culture

The AGS stomach cancer cell line was provided by the Korean Cell Line Bank (Seoul, Republic of Korea). The cells were cultured in RPMI1640 medium supplemented with 10% heat-inactivated fetal bovine serum (FBS; Gibco, Grand Island, NY, USA) and 1% penicillin–streptomycin (10,000 U/mL) in an incubator at 37 °C with humidified air containing 5% CO_2_ (Gibco, Grand Island, NY, USA).

### 2.4. Hyperthermia

AGS cells were seeded in 6-well plates (0.3 × 10⁶ cells) and suspended in 3 mL of medium. The temperature environment was applied by incubating the cells for 30 min in a water bath set to certain temperatures (37 °C or 43 °C). LR was treated to AGS cells an hour prior to temperature control.

### 2.5. MTT Assay

The MTT assay was used to assess cell proliferation after temperature exposure to LR. AGS cells (1 × 10⁴ cells/100 μL) were seeded in 96-well plates and allowed to attach overnight. Each group contained three wells, with the untreated group acting as a control. LR was added to the plates in three concentrations (80, 100, and 120 μg/mL) after the cells had been fixed. The plates were incubated for an hour at 37 °C in a humid atmosphere with 5% CO_2_ and then soaked in a water bath with a temperature controlled at 37 °C or 43 °C for 30 min. After 48 h, each well received 20 μL of MTT (2 mg/mL in PBS) (AMRESCO, Solon, OH, USA) and then was incubated for an additional 2 h. After removing the growth media, the cells were lysed in 100 μL of DMSO. Absorbance was measured at 570 nm using an automated spectrophotometric plate reader. The relative cell viability percentages were normalized to those of untreated controls. The synergistic effects of the medication and HT combination were calculated using Compusyn software (ver.1.0).

### 2.6. Trypan Blue Assay

The vitality of the cells was assessed using a hemocytometer after trypan blue (Sigma-Aldrich, St. Louis, MO, USA) staining (0.4%, 1:1 dilution in the PBS containing the cells). In brief, AGS cells (0.3 × 10⁶) were sown in 6-well plates and then treated to 1 h of LR therapy and heat (30 min). After 24 h of post-treatment incubation, the cells were collected, diluted 1:4 with PBS, stained, and then counted.
Rate of cell survival = viable cell count/total cell count × 100

### 2.7. Morphology Assa

Cell proliferation was measured using a morphological assay. AGS cells were seeded in a 6-well plate at a density of 0.3 × 10⁶ cells per well. The cells were treated with 120 μg/mL LR for 1 h after adhering to the plates and then incubated at 37 °C or 43 °C for 30 min. The cells were examined and photographed under a microscope (CX-40; Olympus, Tokyo, Japan) after 24 h.

### 2.8. Wound-Healing Assay

Cells were plated in a 6-well plate at a density of 0.3 × 10⁶ cells per well and then kept at 37 °C. Once the cells reached confluence, a thin scratch was created in each well using a yellow pipette tip. Images were captured at 0 h using a microscope (CX-40; Olympus, Tokyo, Japan) (0 h). After 24 h, the cells were rinsed with PBS and photographed under a light microscope (24 h).

### 2.9. Colony Formation Assay

A total of 400 cells were seeded into each well of a 6-well plate and then incubated overnight. The cells were incubated for 30 min at 37 °C or 43 °C and then treated with 120 μg/mL LR for an hour. After a week, the cells were stained with crystal violet solution (Sigma-Aldrich, St. Louis, MO, USA) for 10 min at room temperature and then washed with PBS. Colonies were examined under a microscope (CX-40; Olympus, Tokyo, Japan).

### 2.10. Western Blot Analysis

Protein was extracted from AGS cells after indicated treatments. After calculating protein concentration, equal amounts of the SDS-PAGE-separated lysates were transferred onto polyvinylidene difluoride (PVDF) membranes, which were subsequently blocked at room temperature with 1 × TBS containing 0.1% Tween 20 and 5% skim milk. The membranes were incubated at 4 °C overnight with the following primary antibodies: anti-caspase-3, anti-caspase-8, anti-caspase-9, anti-survivin, anti-HSP27, anti-HSP70, anti-HSP90, anti-p-ERK (Thr202/Tyr204), anti-ERK, anti-p-p38 (Thr180/Tyr182), anti-p38, anti-p-JNK (Thr183/Tyr185), anti-JNK (Cell Signaling Technology, Danvers, MA, USA), anti-β-actin, anti- Bcl-xL, anti-Bcl-2, anti-cyclin B1, anti-cyclin D1, anti-MMP9, anti-MMP2, anti-VEGF (Santa Cruz Biotechnology, Inc., Dallas, TEX, USA), anti-HSF1, anti-pHSF1 (Abcam, Inc., Waltham, MA, USA), and anti-cleaved caspase (Genetex, Irvine, CA, USA). The membranes were washed three times before exposure to diluted anti-rabbit or anti-mouse IgG secondary antibodies (Santa Cruz Biotechnology, Inc.) for an hour at room temperature. The blots were washed thrice with 1 × TBS-T buffer for 10 min between each stage. The membranes were identified using enhanced chemiluminescence (Millipore, Billerica, MA, USA).

### 2.11. Apoptosis Assay

Apoptosis was examined by flow cytometry using the annexin V-FITC detection kit (ApoScan kit, Cat. No.: LR-02-100). Briefly, AGS cells (0.3 × 10⁶ cells/well in a 6-well plate) were subjected to LR and HT for 24 h. The cells were collected and stained with annexin V-FITC in 1× cold binding buffer under a light-blocking cover for 15 min at room temperature. Propidium iodide (PI) staining was performed using 1× cold binding buffer after removing the supernatant.

### 2.12. Cell Cycle Analysis

AGS cells (0.3 × 10⁶ cells/well in 6-well plates) were cotreated for 24 h. To measure the cell cycle phase, cells were collected, frozen in 70% ice-cold EtOH for 24 h incubation, washed in 1× cold PBS, and then resuspended in PBS supplemented with 1 mg/mL PI and 10 mg/mL RNase A in a dark environment for 10 min. The cell cycle was analyzed using a flow cytometer.

### 2.13. Analysis of Reactive Oxygen Species (ROS)

A ROS experiment was carried out using 2′,7′-dichlorofluorescin diacetate (Invitrogen^TM^ D399). A 6-well plate containing AGS cells (0.3 × 10⁶ cells per well) was subjected to cotreatment for 4 h. After collecting the cells, 10 μM reagent was added and then incubated at 37 °C for 40 min without light for reaction. ROS levels were measured using flow cytometry.

### 2.14. Statistical Analysis

All numerical values are represented as the mean ± SD. Statistical significance of the data compared with the untreated control was determined using Student’s unpaired *t*-test. * *p* < 0.05, ** *p* < 0.01, and *** *p* < 0.001.

## 3. Results

### 3.1. UPLC-ESI-QTOF-MS/MS Analysis for the Identification of Chemical Components in LR

The ethanolic extract was subjected to UPLC-ESI-QTOF-MS/MS to determine its chemical profile and identify its constituents. Sixteen components were identified: norisoboldine (3), boldine (5), aesculitannin B (6), norboldine (10), linderalactone (12), lindenanolide E (15), and lindenanolide (16). These compounds are reported as the major components of LR (Figure 1) [16,17,18]. The detected peaks are listed in Table 1.

### 3.2. Cotreatment with LR and 43 °C Hyperthermia Synergistically Inhibits AGS Cell Proliferation

The effect of simultaneous treatment with LR and hyperthermia at 37 °C and 43 °C was investigated using MTT assays. When LR was administered at the same dose (120 µg/mL), the combination of LR and 43 °C significantly reduced AGS cell viability to a greater extent than the combination of LR and 37 °C (Figure 2A). The degree of synergy between LR and hyperthermia was determined using a combination index. In Figure 2B, trypan blue staining indicates the statistical significance of the effect of LR treatment, which is notably enhanced under hyperthermic conditions. We also performed an MTT assay to examine the effect of cotreatment on human gastric normal cells (GES-1). The results confirm that hyperthermia and LR cotreatment showed minimal toxicity on GES-1 cells (Figure 2C).

Morphological observations revealed that the cotreatment led to distinct changes in cell shape. These alterations result from the inhibitory effect of the cotreatment on cell growth (Figure 2D). The morphological features, such as cell shrinkage and membrane blebbing, were consistent with the well-known hallmarks of apoptotic cell death, providing further evidence for the induction of apoptosis through the cotreatment approach (Figure 2D). In addition, crystal violet staining of AGS cells showed that colony formation was notably decreased after cotreatment with LR and 43 °C compared with cotreatment with LR and 37 °C (Figure 2E). Furthermore, we demonstrated that cotreatment with LR and hyperthermia suppressed cell migration (Figure 2F). These findings indicate that the combination of LR and hyperthermia exerts an antiproliferative effect on AGS cells.

### 3.3. Cotreatment with LR and 43 °C Hyperthermia Induces Apoptosis in AGS Cells

To elucidate the mechanisms underlying the synergistic effects of LR and hyperthermia, we investigated the expression levels of various factors associated with apoptosis, cell proliferation, metastasis, and angiogenesis. Our results show that treatment with LR at 43 °C dose-dependently increased the expression of activated forms of caspase 3, a key marker of programmed cell death [19,20], whereas this effect was not observed under normothermic conditions (37 °C) (Figure 3A). Additionally, cotreatment with LR and 43 °C decreased the expression of pro-caspase 8 and 9 (Figure 3B). Moreover, cotreatment with LR and 43 °C significantly reduced the expression levels of antiapoptotic members of the B-cell lymphoma (Bcl)-2 family, including Bcl-2, Bcl-xL, and survivin, in a dose-dependent manner (Figure 3B) [21,22]. Furthermore, cotreatment with LR and hyperthermia effectively inhibited the metastatic potential and mitosis of AGS cells by suppressing the expression of Cyclin D1, VEGF, MMP-2, and MMP-9 (Figure 3C) [23,24,25]. These findings support the notion that cotreatment with LR and hyperthermia exerts a potent anticancer effect by modulating multiple cellular pathways.

### 3.4. Cotreatment with LR and 43 °C Hyperthermia Synergistically Induces Apoptosis and Cell Cycle Arrest in AGS Cells

The rate of annexin V-related apoptosis in AGS cells was higher after cotreatment with LR and hyperthermia than after treatment with 43 °C hyperthermia alone or cotreatment with LR and normothermia (Figure 4A). A dose-dependent effect of the cotreatment was evident as the ratio of apoptotic cells increased by nearly 2.34-fold at the highest LR dose (17.28%). Furthermore, flow cytometric analysis revealed that cotreatment with LR and hyperthermia induced cell cycle arrest at the G2/M phase (Figure 4B). Such a finding was supported by the significant reduction in the expression of cyclin B1 in AGS cells treated with LR at 43 °C hyperthermia (Figure 4C). These results suggest that cotreatment with LR and hyperthermia induces apoptosis and cell cycle arrest in AGS cells, which may contribute to the anticancer effect of cotreatment.

### 3.5. Cotreatment with LR and 43 °C Hyperthermia Suppresses HT-Induced Heat Shock Responses

HSPs are molecular chaperones that maintain protein stability, facilitate transport, and transmit signals within cells [26,27,28]. HSPs also play a crucial role in protecting cells from stress and preventing the degradation of severely damaged proteins [29,30]. The expression levels of HSP27, 70, and 90 were increased in the AGS cells treated with 43 °C hyperthermia (Figure 5A). However, LR treatment significantly reduced the expression of HSPs under both normothermic and hyperthermic conditions. HSF1 is a transcription factor that is activated by stress factors, such as heat shock, leading to the synthesis of HSPs. HSF1 is overexpressed in cancer cells and contributes to tumor cell migration, invasion, and proliferation [31,32]. Hyperthermia treatment at 43 °C can induce HSF1 phosphorylation. In this study, we observed that cotreatment with LR inhibited HSF1 phosphorylation, even when AGS cells were exposed to 6 h of hyperthermia (Figure 5B).

### 3.6. Cotreatment with LR and 43 °C Hyperthermia Synergistically Increases ROS Generation and MAPK Phosphorylation in AGS Cells

HSPs regulate the intracellular levels of reactive ROS; thus, a decreased expression of heat shock proteins can increase ROS levels in cancer cells [29]. Accordingly, we examined the underlying mechanism by which cotreatment with LR and hyperthermia induces AGS cell death to determine the role of ROS in the proapoptotic effect of the combination therapy. Flow cytometric analyses (Figure 6A) showed that cotreatment with LR and 43 °C hyperthermia significantly increased ROS levels when compared with LR treatment at 37 °C (panel 3). Next, we pretreated cells with N-acetylcysteine (NAC), a free radical scavenger used as a ROS inhibitor [33]. NAC pretreatment nullified the effects of LR and hyperthermia on ROS generation (Figure 6A). Considering that increased ROS levels can activate MAPKs, which can consequently induce apoptosis [34], we investigated the effect of ROS on MAPK activation. As shown in Figure 6B, cotreatment with LR boosted the phosphorylation of MAPKs, including JNK, p38, and ERK, induced by 43 °C hyperthermia [35].

Figure 6C shows that NAC pretreatment decreased ROS production in the cotreated AGS cells. Then, we determined whether a decrease in ROS levels could mitigate the effects of the cotreatment. As shown in Figure 6D, LR and 43 °C cotreatment synergistically induced apoptosis in AGS cells, whereas NAC pretreatment decreased the population of apoptotic cells from 28.3% to 21.09%. In line with this, we confirmed that NAC pretreatment reversed the expression changes of caspase-3, HSP27, and HSP70 induced by the cotreatment, indicating that the cotreatment effects were partly dependent on ROS production and HSP expression.

## 4. Discussion

Gastric cancer remains a significant global health threat, particularly in Asian countries, where the prevalence and mortality rates are alarmingly high [36,37]. Various treatment methods, including surgery, chemotherapy, and radiation therapy, are available for patients with gastric cancer; however, these methods have limitations, such as adverse side effects, potential for tumor resistance, and patient tolerance [38,39]. Therefore, novel and effective therapeutic strategies must be developed urgently to overcome these limitations and improve patient outcomes.

One approach to overcome these limitations is the use of hyperthermia therapy [40]. Cancer hyperthermia therapy involves the use of heat to target and selectively destroy cancer cells [41,42]. However, cancer hyperthermia therapy has limited efficacy when used alone because it induces a heat shock response that activates HSFs and HSPs. Such defense mechanisms against heat stimulation allow unexpected tumor cell survival and promote tumor resistance to therapy [43].

LR is a natural herb used in traditional medicine owing to its thermogenic effects; it has been used in the treatment of various gastrointestinal disorders, such as indigestion, abdominal pain, and bloating [44,45,46,47]. Given its historical use and known properties, incorporating LR into cancer hyperthermia therapy could help overcome the drawbacks of hyperthermia treatment and enhance its therapeutic efficacy. This study was initiated with the rationale that LR and hyperthermia cotreatment may synergistically enhance the anticancer properties of both modalities. We conducted a series of experiments on cancer cell proliferation and death to verify the effects and underlying mechanisms of action of LR and hyperthermia.

First, we found through MTT assays, morphological observations, crystal violet staining, and cell migration assays that cotreatment with LR and 43 °C hyperthermia synergistically inhibited AGS cell proliferation. These results suggest that the combination of LR and hyperthermia has a potent anticancer effect against AGS cells (Section 3.2).

The mechanisms underlying the synergistic effect induced by cotreatment with LR and hyperthermia were investigated by examining the expression of various factors associated with apoptosis, proliferation, metastasis, and angiogenesis. Apoptosis is the main mechanism of programmed cell death [48,49] and is considered one of the most important targets for anticancer therapy [50,51]. Our results show that cotreatment with LR and 43 °C hyperthermia induced apoptosis in AGS cells by increasing the expression of activated caspase 3 and decreasing the expression of pro-caspase 8 and 9, as well as antiapoptotic Bcl-2 family members (Section 3.3). The apoptotic effect of the cotreatment with LR and hyperthermia may be attributed to cell cycle arrest. Four distinct phases compose the eukaryotic cell cycle: G1, S (synthesis), G2 (interphase), and M (mitosis/cytokinesis) [41,52]. LR and hyperthermia cotreatment significantly promoted annexin V-related apoptosis and cell cycle arrest in the G2/M phase in AGS cells. These results suggest that the combination therapy effectively disrupts the cell cycle and promotes programmed cell death (Section 3.4).

Cells can also be exposed to various environmental stressors, such as cellular development or disease, temperature change, or mechanical stress [53]. Cells frequently activate defense mechanisms in response to stress [54,55]. Particularly, heat stress induces a heat shock response in cells, including cancer cells [56,57]. Thus, cancer cells often gain resistance to hyperthermia. In the present study, cotreatment with LR suppressed the heat (43 °C hyperthermia)-induced heat shock responses by inhibiting HSF1 phosphorylation and reducing the expression of HSP27, HSP70, and HSP90 (Section 3.5), which are crucial for cell survival and stress resistance [58,59].

ROS, a group of highly bioactive molecules, act as key signaling regulators in several cellular functions [54,55]. Although low to moderate levels of ROS are essential to maintain cellular homeostasis, excessive ROS can induce cellular stress and damage proteins, DNA, lipids, and membranes [60,61]. ROS can induce cancer cell death through several pathways. Thus, ROS is considered as a promising therapeutic target for cancer treatment [62,63,64]. In the present study, cotreatment with LR and 43 °C hyperthermia synergistically increased ROS generation and related MAPK phosphorylation in AGS cells, suggesting a potential molecular mechanism underlying the observed synergistic effects (Section 3.6). This result was further confirmed by showing that ROS were required for the effects of the cotreatment. NAC pretreatment abolished the apoptotic effect of LR and 43 °C hyperthermia cotreatment in AGS cells, demonstrating the crucial role of ROS in mediating the observed anticancer effects (Section 3.6).

Our study demonstrates that LR enhances the sensitivity of gastric cancer cells to hyperthermia treatment. This effect can be attributed to the following bioactive compounds in LR: norisoboldine (O3), boldine (O5), aesculitannin B (O6), norboldine (O10), linderalactone (O12), lindenanolide E (O15), and lindenanolide (O16). These compounds exhibit various biological activities that contribute to their synergistic effects. Norisoboldine (O3) and boldine (O5) exert anticancer activities by modulating the expression of genes related to cell cycle progression, apoptosis, and metastasis [65,66,67,68]. Aesculitannin B (O6) and norboldine (O10) exhibit antitumor properties by inducing cell cycle arrest and apoptosis in cancer cells [69,70]. Linderalactone (O12), lindenanolide E (O15), and lindenanolide (O16) also exert cytotoxic effects on various cancer cell lines [71,72,73]. Furthermore, these phytochemicals may potentiate the effects of hyperthermia by modulating the expression of HSPs, which play a crucial role in protecting cells from thermal stress. These compounds can increase the sensitivity of cancer cells to hyperthermia by inhibiting the expression or activity of HSPs. However, further investigation is warranted to completely elucidate the hyperthermia-potentiating effects of the components of LR.

Despite these promising results, this study has several limitations, mainly due to its in vitro nature. Further research is required to confirm these findings in animal models and human clinical trials. A number of attempts have been made to demonstrate the effect of hyperthermia in vivo. One of the most easily integrated and convenient methods involves using magnetic nanoparticles [74]. However, clinical application of nanoparticles is still questionable, so instead we are seeking a method that resembles the function of clinically available hyperthermia devices. At the moment, it is difficult to share all the details of what we are designing, but we are mainly referring to two previous studies: one equipped with a microwave hyperthermia system [75] and another demonstrating a magnetic field method [76]. So, to address the gap between our study and clinical applications, we are currently engaged in collaborative research with engineers to develop a sophisticated animal experimental model to provide translational evidence. Our approach goes beyond simply raising the temperature in animal cages; we are planning to attach magnetically responsive materials to tumor masses, enabling targeted and effective hyperthermia treatment. This novel methodology forms the cornerstone of our future investigations, aiming to elucidate the precise molecular mechanisms underlying the observed synergistic effects, and to determine the optimal dosage and duration of the combined treatment. The anticipated in vivo studies will not only validate the in vitro findings but will also pave the way for potential clinical applications, enhancing our understanding of this promising therapeutic strategy.

In conclusion, our study provides evidence that cotreatment with LR and hyperthermia exerts a synergistic anticancer effect on AGS cells by inducing apoptosis, promoting cell cycle arrest, and suppressing heat-induced heat shock responses. Our research endeavors to lay the groundwork for the development of groundbreaking and more efficient therapeutic strategies for patients with gastric cancer, ultimately enhancing their quality of life and survival rates. Future investigations should focus on validating these results in vivo, unraveling the molecular pathways underlying the observed effects, and refining treatment protocols.

## Figures and Tables

**Figure 1 biomedicines-11-02710-f001:**
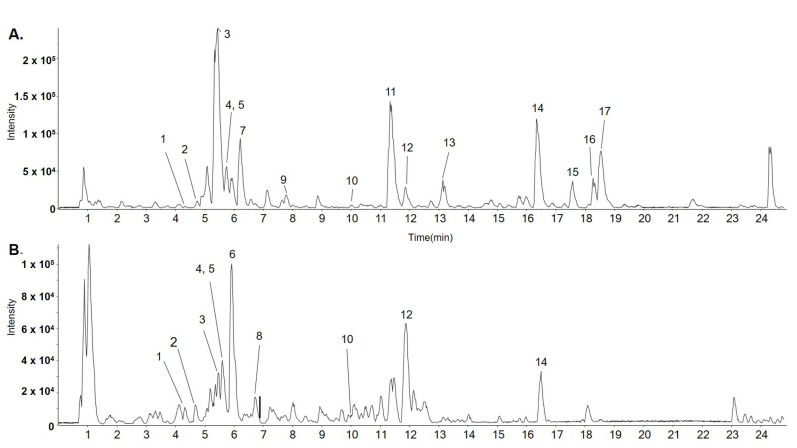
Representative base peak chromatogram (BPC) identified in LR using LC-ESI-QTOF MS/MS analysis in positive (**A**) and negative ion modes (**B**).

**Figure 2 biomedicines-11-02710-f002:**
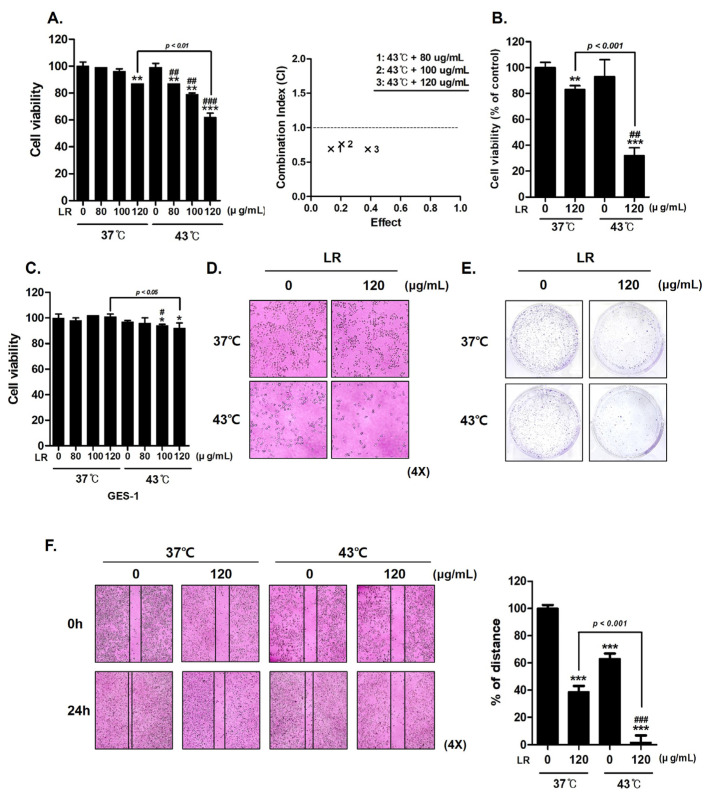
Effect of combined LR and hyperthermia on AGS cell viability. AGS cells were exposed to various concentrations of LR (0, 80, 100, and 120 μg/mL) with or without hyperthermia at 43 °C for 24 h. (**A**) MTT assay was used to calculate the percentage of cell viability, and Compusyn software was used to calculate the combination index. (**B**) Cell viability under cotreatment with LR and hyperthermia was compared with that under normothermia by performing a trypan blue assay. (**C**) MTT assay was used to investigate the effect of cotreatment on GES-1, human gastric normal cells. (**D**) Morphological changes indicating apoptosis were observed under a regular light microscope. (**E**) Crystal violet staining was used for the clonogenic experiment. (**F**) Wound-healing assay was conducted. * *p* < 0.05, ** *p* < 0.01, and *** *p* < 0.001 vs. control group; # *p* < 0.05, ## *p* < 0.01, and ### *p* < 0.001 vs. 43 °C + 0 μg/mL group.

**Figure 3 biomedicines-11-02710-f003:**
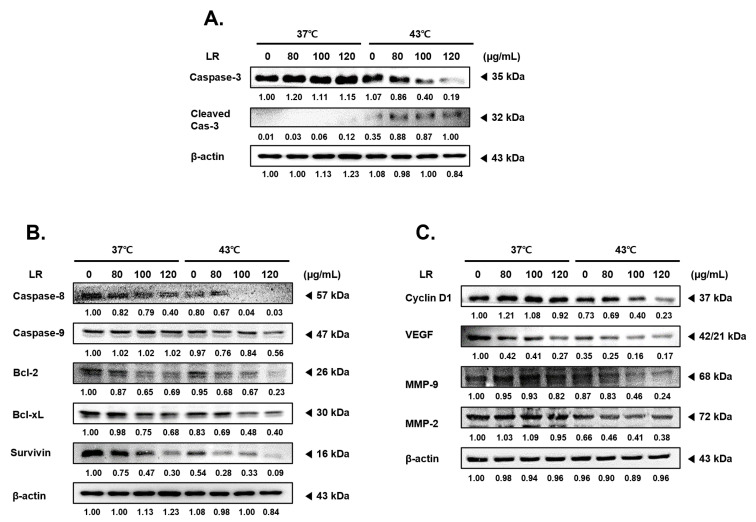
Effects of LR combination with hyperthermia on the expression levels of angiogenesis, survival, and proliferation in the treatment and control groups. LR was applied to AGS cells (0.3 × 10⁶ cells) with or without hyperthermia, and the cells were incubated for 24 h. Equal volumes of lysates from whole-cell extracts were then subjected to Western blot analysis. Western blot assays were used to determine the protein expression of (**A**) caspase-3; (**B**) caspase-8, caspase-9, Bcl-2, Bcl-xL, and Survivin; and (**C**) Cyclin D1, VEGF, MMP-9, and MMP-2. β-actin was used as a loading control.

**Figure 4 biomedicines-11-02710-f004:**
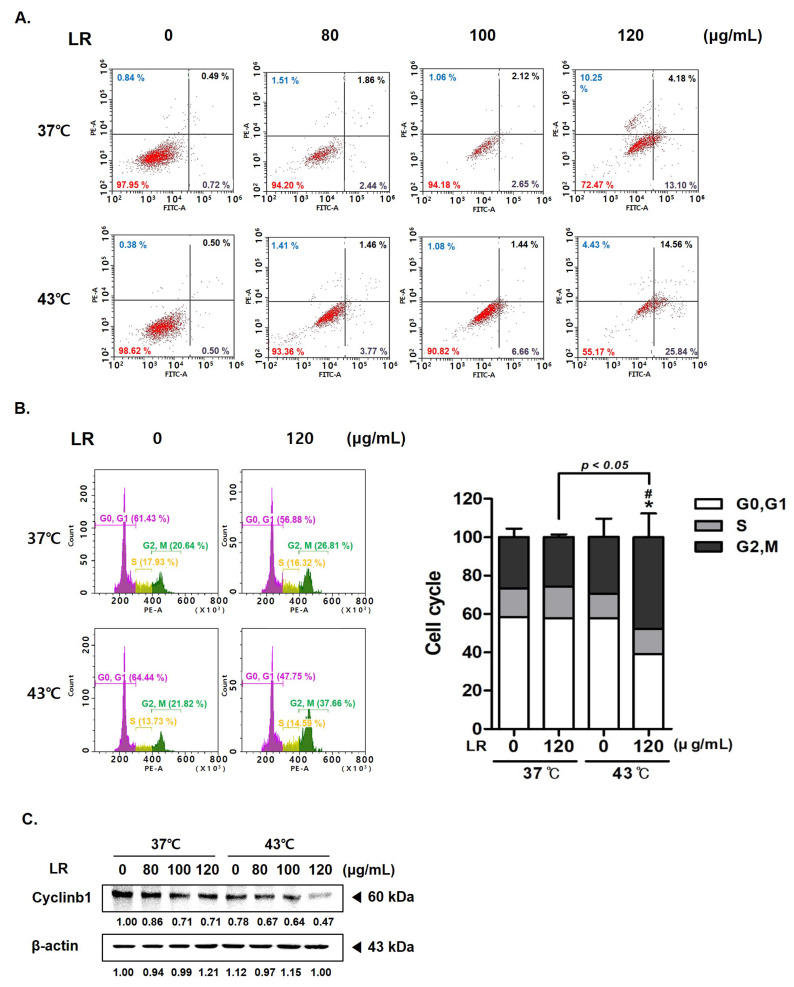
Effect of LR and hyperthermia combination on the apoptosis and cell cycle in AGS cells. AGS cells (0.3 × 10⁶ cells) were treated with LR (0 or 120 μg/mL) with or without hyperthermia. Apoptotic cells were detected through annexin V and PI staining and then analyzed using a flow cytometer. Flow cytometric analysis on (**A**) apoptosis profile and (**B**) cell cycle profile was performed. (**C**) Cyclin B1 expression was measured through Western blot assay. β-actin was used as a loading control. ** p* < 0.05 vs. control group; # *p* < 0.05 vs. 43 °C + 0 μg/mL group.

**Figure 5 biomedicines-11-02710-f005:**
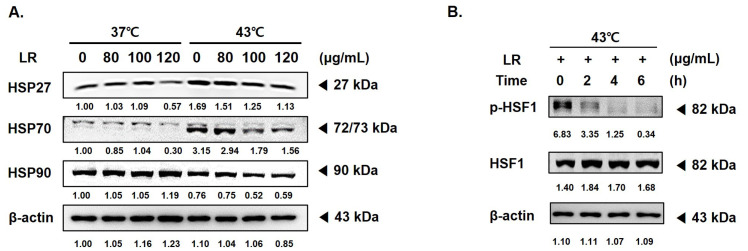
Effect of LR and hyperthermia cotreatment on HSP and ROS correlation in AGS cells. AGS cells (0.3 × 10⁶ cells) were treated with LR (0 or 120 μg/mL) with or without hyperthermia. Protein expression of (**A**) HSP27, HSP70, and HSP90 and (**B**) p-HSF1 and HSF were assessed using Western blot. β-actin was used as a loading control.

**Figure 6 biomedicines-11-02710-f006:**
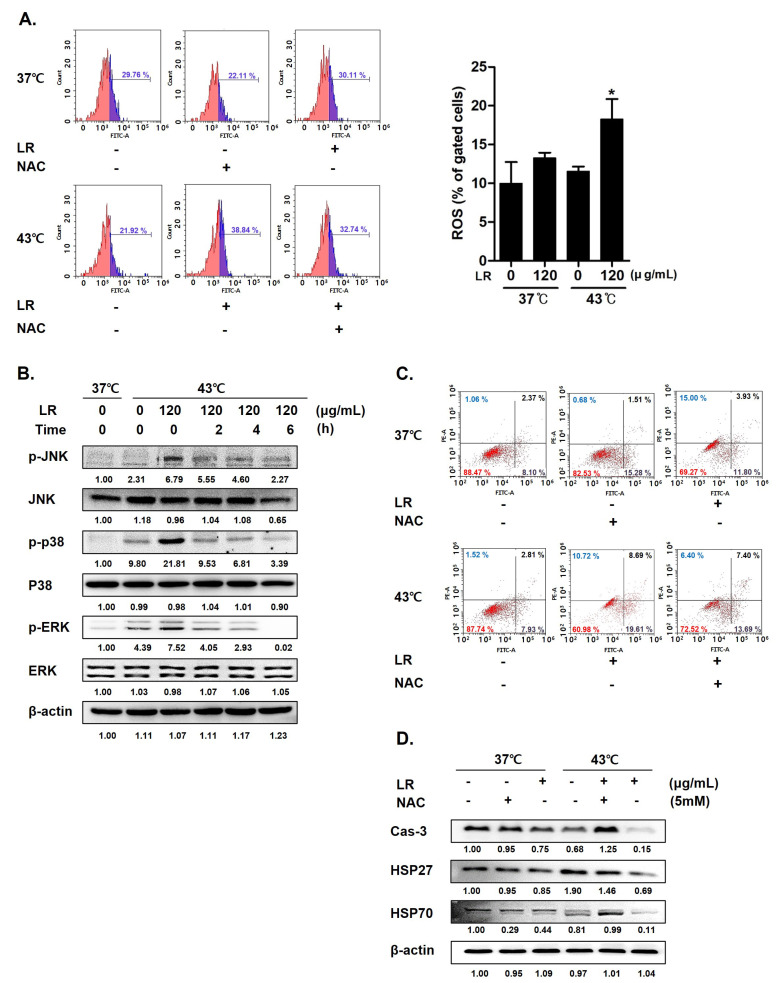
Effect of LR and hyperthermia on ROS generation, apoptotic markers, and MAPK pathway in ROS-inhibited AGS cells. Before being exposed to LR (0 or 120 μg/mL) with or without hyperthermia at 43 °C, AGS cells were pretreated with N-acetylcysteine (NAC, 5 mM) for 1 h. (**A**) ROS production was examined using flow cytometry. (**B**) Western blot was used to determine the levels of p-JNK, JNK, p-p38, p38, p-ERK, and ERK. (**C**) Apoptosis profiling was performed using flow cytometry. (**D**) Caspase-3, HSP27, and HSP70 protein expressions. (−), absence of NAC or LR; (+), presence of NAC or LR. ** p* < 0.05 vs. control group.

**Table 1 biomedicines-11-02710-t001:** Detected peak list from the UPLC-ESI-QTOF-MS/MS analysis of LR.

No.	Name	Formula	Mass (Da)	Expected RT (min)	Adduct	Found at Mass (Da)	Error (ppm)	MS/MS Product Ions	Identified with	Peak Area
1	Epigallocatechin	C_15_H_14_O_7_	306.0740	4.33	[M + H]^+^	307.0810	−0.8	139.0386, 163.0389, 135.0429, 177.0540	^#^	2327
					[M − H]_−_	305.0667	0.0	125.0250, 137.0248, 139.0401,165.0194,167.0346		7248
2	Catechin	C_15_H_14_O_6_	290.0790	4.72	[M + H]^+^	291.0864	0.2	139.0386, 123.0440, 147.0438, 207.0648	^#^	2764
					[M − H]_−_	289.0717	−0.1	203.0714, 123.0456, 109.0305, 151.0402, 125.0250		7820
3	Norisoboldine	C_18_H_19_NO_4_	313.1314	5.41	[M + H]^+^	314.1390	0.9	237.0918, 265.0864, 205.0655, 297.1130	^#^	247,192
					[M − H]_−_	312.1243	0.6	297.1015, 282.0772, 254.0824, 239.0711		17,959
4	Epicatechin	C_15_H_14_O_6_	290.0790	5.68	[M + H]^+^	291.0864	0.3	139.0394, 123.0453, 147.0444, 207.0646	^#^	3693
					[M − H]_−_	289.0717	−0.4	203.0721, 123.0453, 245.0827, 109.0307		15,349
5	Boldine	C_19_H_21_NO_4_	327.1471	5.71	[M + H]^+^	328.1545	0.6	265.0862, 237.0913, 297.1122, 205.0651, 177.0697	^#^	41,973
					[M − H]_−_	326.1398	0.0	311.1156, 296.0934, 268.0732, 239.0688		1958
6	Aesculitannin B	C_45_H_36_O_18_	864.1902	5.92	[M − H]_−_	863.1838	1.1	411.0719, 711.1382, 289.0714, 451.1036	*	65,301
7	Reticuline	C_19_H_23_NO_4_	329.1627	6.21	[M + H]^+^	330.1699	−0.1	192.1026, 137.0601, 143.0494, 175.0757	^#^	26,257
8	Lyoniresinol 3a-O-b-D-glucopyranoside	C_28_H_38_O_13_	582.2312	6.71	[M − H]_−_	581.2239	0.0	419.1696, 404.1477, 371.1115, 401.1591	*	7223
9	Alangionoside L	C_19_H_32_O_7_	372.2148	7.70	[M + H]^+^	373.2219	−0.5	175.1482, 133.1017, 119.0864, 193.1583	*	2927
10	Norboldine	C_18_H_19_NO_4_	313.1314	10.00	[M + H]^+^	314.1386	−0.3	177.0544, 145.0283, 121.0650, 89.0396	*	3697
					[M − H]_−_	312.1241	−0.2	148.0532, 178.0501, 190.0508, 297.1008		3645
11	Unknown	C_21_H_28_N_2_O	324.2202	11.37	[M + H]^+^	325.2276	0.5	91.0556, 86.0980, 233.1655, 84.0824	*	125,079
12	Linderalactone	C_15_H_16_O_3_	244.1099	11.88	[M + H]^+^	245.1171	−0.5	141.0700, 156.0934, 165.0698, 105.0702, 91.0552	*	20,652
					[M − H]_−_	243.1026	−0.4	183.0811, 199.1122, 197.0965, 182.0727, 155.0859		19,844
13	Hydroxylindestenolide or linderanolide G	C_15_H_18_O_3_	246.1256	13.15	[M + H]^+^	247.1330	0.4	91.0561, 107.0868, 153.0699, 168.0931, 141.0701	*	28,925
14	Isolinderalactone	C_15_H_16_O_3_	244.1099	16.48	[M + H]^+^	245.1172	0.1	199.1111, 141.0696, 156.0928, 143.0853, 165.0694	*	85,662
					[M − H]_−_	243.1028	−0.3	183.0815, 199.1121, 182.0739, 130.9660		20,421
15	Lindenanolide E or Linderane	C_15_H_16_O_4_	260.1049	17.57	[M + H]^+^	261.1121	0.0	173.0955, 145.1007, 158.0725, 129.0696, 130.0779	*	24,396
16	Lindenanolide H	C_17_H_20_O_4_	288.1362	18.20	[M + H]^+^	289.1433	0.2	155.0854, 183.1164, 168.0937, 229.1229	*	4131
17	Unknown	C_15_H_16_O	212.1201	18.53	[M + H]^+^	213.1273	−0.3	165.0705, 128.0624, 141.0704, 155.0859, 180.0933	*	68,194

# In-house ms/ms library and online database; such as GNPS, MASS bank, or Metlin. * Extract MS with isotope mass.

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
