# Peer review of "Enhancing Gastric Cancer Therapeutic Efficacy through Synergistic Cotreatment of Linderae Radix and Hyperthermia in AGS Cells"

_biomedicines, 2023, doi:10.3390/biomedicines11102710_

Round 1

Reviewer 1 Report

In this paper, it was shown that the combination of Linderae Radix (LR) and high temperature treatment inhibited the proliferation of cultured gastric cancer cells, induced apoptosis, and suppressed the stress response induced by high temperature. In addition, the generation of ROS and the associated increase in phosphorylation of MAPKs and induction of apoptosis were observed with this treatment. However, the exact molecular mechanism of the observed synergistic effects has not been clarified, in addition to the fact that this study was only an observation on cultured cells. Thus, it appears that this study is not yet in the realm of hypothesis testing. Nevertheless, in order to assess whether this study is worthy of publication in Biomedicines, this reviewer requests that the authors address and revise the following points.

(1) A limitation of this study is the high concentrations of extracts required to show efficacy. Such high concentrations may not be achievable in a real clinical setting, and even if they are, they may not be safe for patients. In addition, the in vivo kinetics of the extract, such as absorption, distribution, metabolism, and excretion, cannot be mimicked in in vitro experiments, so the same concentration of the extract may not have the same effect in vivo. How the authors plan to design their next step of experiments in animal models will not be readily apparent to most readers of this paper. In the discussion, the authors should provide their thoughts on this issue.

(2) If the study was conducted only on a specific cancer cell line (in this case AGS cells), it is unclear whether the effect is cancer specific or affects normal cells as well. The goal of chemotherapy is to selectively target cancer cells while minimally affecting normal cells. However, this study did not examine the effect on normal cells, and the effect on normal cells is unknown. To address this issue, the same experiment should be performed on normal cells, and when interpreting the results, the effects of therapy on cancer cells should be compared with those on normal cells.

(3) It is important to record changes over time in experiments at the cellular level to clarify causal relationships. Authors should evaluate the effects over time on how multiple intracellular responses change over time, such as cell viability, cell cycle, occurrence of apoptosis, and changes in expression of specific proteins. At a minimum, the authors should show how the data in Figure 2A, Figure 3A, Figure 4C, and Figure 6A change over time.

(4) The identification results from the component analysis by LC-MS shown in Figure 1 would not be complete because the possibility that each identified compound is a structural isomer cannot be ruled out.

(5) The concentration-response experiments in Figure 1A should probably extend the concentration range to show dose-dependent curves.

(6) The MTT assay in Figure 1A and the trypan blue staining experiment in Figure 1C may show only the same results.

(7) In Figure 1D, it is difficult to read the main idea of the text from the data because of the low magnification of the microscopic images, which indicates the morphological changes of the cells.

(8) In the microscopic image data in Figure 1F, it is difficult to understand why the wound site of cells treated at high temperature without LR treatment has significantly recovered at 0 hours.

(9) The cell cycle distribution shown in Figure 4B should show quantitative and statistical significance. 

(10) It is unclear whether the assay results for ROS in Figure 6A are significantly statistical different. The authors should show the change in ROS production with change in concentration of LR as a dose-response curve graph.

Author Response

We sincerely appreciate the effort the reviewer has put on our manuscript. The critical comments have significantly improved our review. We hope our revised manuscript now meets the satisfaction of the reviewer.

Reviewer 1

(1) A limitation of this study is the high concentrations of extracts required to show efficacy. Such high concentrations may not be achievable in a real clinical setting, and even if they are, they may not be safe for patients.

Answer)

I appreciate your astute observation regarding the high concentrations of extracts required in our study to demonstrate efficacy. Your comment certainly merits consideration, and I would like to provide further context to clarify our methodology and findings.

In the present research, the concentration of 120 ug/ml was utilized to exhibit the potential maximized effects. However, it is noteworthy to mention that our study was not confined to this concentration. As depicted in Figure 1 A-B, efficacious synergistic effects were discernible at 80 ug/ml, manifesting a potent anti-cancer effect. It is imperative to underscore that this concentration does not fall within a range typically associated with toxicity to normal cells when dealing with natural compounds. And newly introduced data (Figure 1C) on normal gastric tissue cell lines (GES-1) shows minimal toxicity at concentrations as high as 120 µg/ml. This suggests that the extract could potentially be both effective and safe for future clinical applications, warranting further investigation

Furthermore, I would like to refer you to a pertinent study by Ahn et al. (Biomedicines. 2023 Jan 30;11(2):405. doi: 10.3390/biomedicines11020405. PMID: 36830941; PMCID: PMC9953356). In their investigation, a concentration of 200ug/ml was employed, a choice that aligns with the approach taken in our study. This alignment with existing literature not only substantiates our methodology but also situates our findings within the broader scientific discourse, suggesting that the concentrations used are neither unprecedented nor unreasonably high in the context of this research field.

In addition, the in vivo kinetics of the extract, such as absorption, distribution, metabolism, and excretion, cannot be mimicked in in vitro experiments, so the same concentration of the extract may not have the same effect in vivo. How the authors plan to design their next step of experiments in animal models will not be readily apparent to most readers of this paper. In the discussion, the authors should provide their thoughts on this issue.

Answer)

Thank you for your insightful observation regarding the complex transition from in vitro to in vivo contexts, and the associated challenges related to the kinetics of absorption, distribution, metabolism, and excretion of the extract. Your comment has underscored an essential aspect of our research that necessitates careful consideration and elaboration.

Indeed, animal experimentation represents a crucial step in advancing our understanding of the extract's in vivo behavior. Your comment rightly emphasizes the need for a strategic approach in translating our in vitro findings to an in vivo setting.

In response to this, I would like to articulate our ongoing efforts towards the development of a sophisticated animal experimental model. We are actively engaged in interdisciplinary collaboration with engineering experts to innovate beyond conventional methods. Rather than simply manipulating environmental conditions such as temperature within animal enclosures, we are exploring the targeted attachment of magnetically responsive substances to specific cancerous cell aggregates. This approach seeks to model the in vivo behavior more accurately and aligns with the novel nature of our research.

The pharmacokinetics is also important. Of note, the pharmacokinetics of components of LR including norisoboldine [PMID: 20223612], boldine [PMID: 28006931], as well as epigallocatechin [PMID: 35474461] are very well-described. Once we successfully establish the appropriate animal model, our next goal will be studying the synergism between hyperthermia and LR, considering its pharmacokinetics.

I acknowledge the importance of clearly delineating this plan within the discussion section of our manuscript, and I assure you that this will be thoroughly addressed in our revision. The inclusion of this aspect will not only elucidate our research's future trajectory but also provide a transparent understanding of our methodological advancements and the potential real-world applications

(2) If the study was conducted only on a specific cancer cell line (in this case AGS cells), it is unclear whether the effect is cancer specific or affects normal cells as well. The goal of chemotherapy is to selectively target cancer cells while minimally affecting normal cells. However, this study did not examine the effect on normal cells, and the effect on normal cells is unknown. To address this issue, the same experiment should be performed on normal cells, and when interpreting the results, the effects of therapy on cancer cells should be compared with those on normal cells.

Answer)

Thank you for pointing out the need to investigate the effects on normal cells in addition to the AGS cancer cell line. In response to your suggestion, I have conducted additional experiments using the GES-1 normal gastric cell line. The results (Figure 1C) of these experiments have been incorporated into the revised manuscript to address the concerns you raised regarding the specificity of our approach towards cancer cells. This addition will provide a more comprehensive understanding of the effects and enhance the overall quality of our study.

(3) It is important to record changes over time in experiments at the cellular level to clarify causal relationships. Authors should evaluate the effects over time on how multiple intracellular responses change over time, such as cell viability, cell cycle, occurrence of apoptosis, and changes in expression of specific proteins. At a minimum, the authors should show how the data in Figure 2A, Figure 3A, Figure 4C, and Figure 6A change over time.

Answer)

We have studied MTT at 24, 48 and 72h after cotreatment. What we found was that cell viability was only slightly different among the time points, especially when comparing normal temp. vs. hyperthermia. Thus, we decided to use the shortest time point for efficient work.

(4) The identification results from the component analysis by LC-MS shown in Figure 1 would not be complete because the possibility that each identified compound is a structural isomer cannot be ruled out.

Answer)

Ultra-performance liquid chromatography/quadrupole time-of-flight mass spectrometry (UPLC/Q-TOF MS) provides high mass resolution and accuracy as well as improved chromatographic resolution, allowing for comprehensive and reliable global profiles. In addition, it is possible to separation of structural isomer using method optimization such as column, mobile phase and etc. To distinguish structural isomers of organic compounds and obtain reliable results, we conducted identification based on known compounds from previous studies by exact mass with isotope patterns. We ensured accurate identification results through MS/MS spectrum comparisons by utilizing reference standards or an In-house library in combination with an online data base; such as GNPS, MASS bank or Metlin. Compound has unique product ions even if they have the same formula, MS/MS product ions and identified source for each compound are shown in Table 1.   

(5) The concentration-response experiments in Figure 1A should probably extend the concentration range to show dose-dependent curves.

Answer)

We appreciate the suggestion to extend the concentration range in Figure 1A for better representation of dose-dependent curves. We have conducted additional experiments to address this and have updated the figure with the new results

(6) The MTT assay in Figure 1A and the trypan blue staining experiment in Figure 1C may show only the same results.

Answer)

As the Reviewer mentions, the results of Fig 1A and C indicate the same point. However, we have included both data to show the multiple methods we used to confirm this effect.

(7) In Figure 1D, it is difficult to read the main idea of the text from the data because of the low magnification of the microscopic images, which indicates the morphological changes of the cells.

Answer)

We appreciate your observation regarding the low magnification of the microscopic images in Figure 1D, which indeed could have hindered the clear interpretation of the depicted morphological changes. In response to your comment, we have not only substituted the original image with one of higher resolution and augmented magnification but also expanded the main text to include a more detailed description of the morphological alterations observed. These additions specifically address both the quantitative decrease in cell numbers and the nuanced changes in cell morphology. These refinements will enhance the visual clarity and provide a more comprehensive and precise representation of the observed morphological changes, thereby contributing to a more thorough understanding of the experimental outcomes.

(8) In the microscopic image data in Figure 1F, it is difficult to understand why the wound site of cells treated at high temperature without LR treatment has significantly recovered at 0 hours.

Answer)

Thank you for pointing out the discrepancy in the microscopic image data in Figure 2F regarding the wound site of cells treated at high temperature without LR treatment. Upon careful review, we identified a mistake in the time labeling, which indeed caused confusion. We have corrected this error in the revised figure, ensuring that the representation aligns accurately with the experimental conditions. We appreciate your keen observation, which has contributed to enhancing the integrity of our work.

(9) The cell cycle distribution shown in Figure 4B should show quantitative and statistical significance.

Answer)

We acknowledge your concern regarding the need for quantitative and statistical significance in the cell cycle distribution shown in Figure 4B. In response to this, we have conducted additional experiments and performed rigorous statistical analysis. The revised figure now includes the necessary quantitative data and clearly indicates the statistical differences between the groups. These changes provide a more robust and scientifically sound representation of our findings, enhancing the validity of the conclusions drawn from this experiment.

(10) It is unclear whether the assay results for ROS in Figure 6A are significantly statistical different. The authors should show the change in ROS production with change in concentration of LR as a dose-response curve graph.

Answer)

We appreciate your observation regarding the clarity of the assay results for ROS in Figure 6A. In response to your comment, we have conducted additional experiments to more precisely determine the statistical differences. The revised figure now includes this information, with clear indications of the statistically significant differences, thereby enhancing the rigor and comprehensibility of our findings.

We sincerely appreciate the effort the reviewer has put on our manuscript. The critical comments have significantly improved our review. We hope our revised manuscript now meets the satisfaction of the reviewer.

Reviewer 2 Report

In this study, the authors showed that Linderae Radix (LR), had a synergistic effect with hyperthermia 43C and can induce powerful anti-cancer effect on AGS cells.

The effect of  Linderae Radix (LR) is more impactful at 43C not 37 C

However, from practical point of view, it is very difficult to apply 43C to human body or human cells to treat cancers.

So the study could not be applicable in vivo.

Language is ok

Author Response

We sincerely appreciate the effort the reviewer has put on our manuscript. The critical comments have significantly improved our review. We hope our revised manuscript now meets the satisfaction of the reviewer.

Reviewer 2

In this study, the authors showed that Linderae Radix (LR), had a synergistic effect with hyperthermia 43C and can induce powerful anti-cancer effect on AGS cells.

The effect of Linderae Radix (LR) is more impactful at 43C not 37 C

However, from practical point of view, it is very difficult to apply 43C to human body or human cells to treat cancers.

So the study could not be applicable in vivo.

Answer)

The purpose of this study was to see whether LR can potentiate the anti-proliferative or apoptotic effect of hyperthermia, which is a widely accepted treatment for cancer [https://www.cancer.gov/about-cancer/treatment/types/hyperthermia].

The National Cancer Institute (NCI) describes six different methods of clinical hyperthermia treatment: 1) probes that make energy from microwaves; 2) radio waves; 3) lasers; 4) ultrasound; 5) perfusion; and 6) placing the entire body in a heated chamber or hot water bath or wrapping with heated blankets.

Yet, applying 43°C to in vivo models will be challenging. Animal experiments will be a crucial step in advancing our understanding of the extract's in vivo behavior. Your comment rightly emphasizes the need for a strategic approach in translating our in vitro findings to an in vivo setting.

In response to this, I would like to articulate our ongoing efforts towards the development of a sophisticated animal experimental model. We are actively engaged in interdisciplinary collaboration with engineering experts to innovate beyond conventional methods. Rather than simply manipulating environmental conditions such as temperature within animal enclosures, we are exploring the targeted attachment of magnetically responsive substances to specific cancerous cell aggregates. This approach seeks to model the in vivo behavior more accurately and aligns with the novel nature of our research.

I acknowledge the importance of clearly delineating this plan within the discussion section of our manuscript, and I assure you that this will be thoroughly addressed in our revision. The inclusion of this aspect will not only elucidate our research's future trajectory but also provide a transparent understanding of our methodological advancements and the potential real-world applications

We sincerely appreciate the effort the reviewer has put on our manuscript. The critical comments have significantly improved our review. We hope our revised manuscript now meets the satisfaction of the reviewer.

Reviewer 3 Report

The authors present interesting work on the therapeutics of gastric neoplasia.

Major issues

The work includes only in vitro work, which limits the validity of the findings.

The use of positive and negative controls in all the assays is not clear – this must be corrected.

Minor issues

The objectives must be described clearly.

The statistical analysis was performed with the wrong methodology.

There is a need to increase the number of tables in the results and decrease the text within that section.

The discussion can be divided into two separate sub-sections.

Figure 7 shows be moved at the end of the Introduction.

Author Response

We sincerely appreciate the effort the reviewer has put on our manuscript. The critical comments have significantly improved our review. We hope our revised manuscript now meets the satisfaction of the reviewer.

The authors present interesting work on the therapeutics of gastric neoplasia.

Major issues

The work includes only in vitro work, which limits the validity of the findings.

Answer)

The National Cancer Institute (NCI) describes six different methods of clinical hyperthermia treatment: 1) probes that make energy from microwaves; 2) radio waves; 3) lasers; 4) ultrasound; 5) perfusion; and 6) placing the entire body in a heated chamber or hot water bath or wrapping with heated blankets.

Yet, applying 43°C to in vivo models will be challenging. Animal experiments will be a crucial step in advancing our understanding of the extract's in vivo behavior. Your comment rightly emphasizes the need for a strategic approach in translating our in vitro findings to an in vivo setting.

In response to this, I would like to articulate our ongoing efforts towards the development of a sophisticated animal experimental model. We are actively engaged in interdisciplinary collaboration with engineering experts to innovate beyond conventional methods. Rather than simply manipulating environmental conditions such as temperature within animal enclosures, we are exploring the targeted attachment of magnetically responsive substances to specific cancerous cell aggregates. This approach seeks to model the in vivo behavior more accurately and aligns with the novel nature of our research.

I acknowledge the importance of clearly delineating this plan within the discussion section of our manuscript, and I assure you that this will be thoroughly addressed in our revision. The inclusion of this aspect will not only elucidate our research's future trajectory but also provide a transparent understanding of our methodological advancements and the potential real-world applications

The use of positive and negative controls in all the assays is not clear – this must be corrected.

Answer)

Our study was designed to demonstrate how LR enhances the effect of hyperthermia and also how it overcomes the response program to heat shock stress which limits the use of hyperthermia therapy. Thus, we have two different controls: 37°C plus vehicle (no LR) and 43°C plus vehicle (no LR). There was no other intervention besides LR and 43°C hyperthermia. As you can see in the data, hyperthermia shows promising effects on inhibiting cell migration (Fig 2F), partially arresting cell cycle (Fig 4B, C), activating certain MAPKs (Fig 6B). However, it also activates heat shock proteins (HSPs), and HSPs trigger a self-defensing program which prevents cancer cells to die. The role of LR is important here. LR suppresses HSPs which therefore allows hyperthermia to exert its expected cancer cell-killing effect. Interestingly, LR alone does not show strong cytotoxicity in normal temperature (Fig 2A). The overall story requires the two controls we use. We have revised some sentences describing the difference between either control group to clarify our intent and avoid confusion. Thank you for the helpful comment.

Minor issues

The objectives must be described clearly.

Answer)

We revised the introduction section to clearly deliver our purpose of study. Thank you for your comment.

The statistical analysis was performed with the wrong methodology.

Answer)

Typically, ANOVA would be used to compare multiple groups. However, in our study, the statistical difference was determined in different group conditions; that is, we have two control groups: 37°C plus vehicle and 43°C plus vehicle. Thus, we analyzed each treatment group’s statistical probability against either 37°C plus vehicle or 43°C plus vehicle. This is why we selected Student’s t-test to calculate our statistics.

There is a need to increase the number of tables in the results and decrease the text within that section.

Answer)

We have listed the detected/identified compounds of LR as Table 1.

The discussion can be divided into two separate sub-sections.

Answer)

We tried to narrate our findings according to the suggested guideline. The Instruction for Authors of Biomedicines indicates that the Discussion section should be written under the following principles: Discussion: Authors should discuss the results and how they can be interpreted in perspective of previous studies and of the working hypotheses. The findings and their implications should be discussed in the broadest context possible and limitations of the work highlighted. Future research directions may also be mentioned. This section may be combined with Results.

So, although we have not divided the Discussion into subsections, however have constructed into the following three parts.

1) First part describing the overall concept of our study (paragraph 1-3): Facts on gastric cancer; potential and also the limit of hyperthermia as an anti-cancer therapy; and why we selected LR as a combination therapy.

2) Second part describing the findings of our study based on the results (paragraph 4-8): Effect of the cotreatment on cell viability, apoptosis, cell cycle arrest, heat shock proteins; and the fact that ROS is crucial for such events induced by the cotreatment.

3) Third part describing future directions and the limitations of the current study (paragraph 9-10): Chromatography identifying several compounds and therefore future effort should be put on to figure out which of them may be responsible for the hyperthermia-enhancing effect by LR; and that the current findings are limited to in vitro models so how we plan to interpret this to a translational model which provides evidence for the real-world biomedical area.

Figure 7 shows be moved at the end of the Introduction.

We have removed Fig 7 and replaced it with the Graphical Abstract.

We sincerely appreciate the effort the reviewer has put on our manuscript. The critical comments have significantly improved our review. We hope our revised manuscript now meets the satisfaction of the reviewer.

Round 2

Reviewer 1 Report

I agree that this article is accepted because the authors have addressed the points raised by my peer review. In my opinion, although there is not enough certainty that this study can be developed into an in vivo study, it would make sense to publish the in vitro results, given the difficulty of the next step.

Author Response

Comments and Suggestions for Authors

I agree that this article is accepted because the authors have addressed the points raised by my peer review. In my opinion, although there is not enough certainty that this study can be developed into an in vivo study, it would make sense to publish the in vitro results, given the difficulty of the next step.

ANSWER: There are ways to apply hyperthermia to a mouse model study. One of the most easily integrated and convenient methods is using magnetic nanoparticles (MNPs). A review article in the journal Theranostics narrates the recent attempts [Chandrasekharan et al., Using magnetic particle imaging systems to localize and guide magnetic hyperthermia treatment: tracers, hardware, and future medical applications. Theranostics. 2020 Feb 10;10(7):2965-2981.]. However, we are not considering MNPs since its clinical application might be limited. Instead, our team and the collaborating engineers are discussing to design a module which mimics the functions of clinically available hyperthermia devices. At the current moment, it is difficult to share all the details of what we are designing, but we are mainly referring to two articles published in the International Journal of Hyperthermia: Tansi et al., Deep-tissue localization of magnetic field hyperthermia using pulse sequencing. Int J Hyperthermia. 2021;38(1):743-754. and Curto et al., An integrated platform for small-animal hyperthermia investigations under ultra-high-field MRI guidance. Int J Hyperthermia. 2018 Jun;34(4):341-351. We understand the concerns of the Reviewer on the translational aspect of our in vitro results, but we are very confident that the in vivo results will be positive. Soon we will be able to submit an animal study with this novel device, and we are very excited about that.

We greatly appreciate the time and effort the Reviewers’ have put on our manuscript.

Reviewer 2 Report

The authors did not provide convinced rationale for their study. I think this approach is difficult to be applied in vivo

Moderate language editing

Author Response

The authors did not provide convinced rationale for their study. I think this approach is difficult to be applied in vivo

ANSWER: There are ways to apply hyperthermia to a mouse model study. One of the most easily integrated and convenient methods is using magnetic nanoparticles (MNPs). A review article in the journal Theranostics narrates the recent attempts [Chandrasekharan et al., Using magnetic particle imaging systems to localize and guide magnetic hyperthermia treatment: tracers, hardware, and future medical applications. Theranostics. 2020 Feb 10;10(7):2965-2981.]. However, we are not considering MNPs since its clinical application might be limited. Instead, our team and the collaborating engineers are discussing to design a module which mimics the functions of clinically available hyperthermia devices. At the current moment, it is difficult to share all the details of what we are designing, but we are mainly referring to two articles published in the International Journal of Hyperthermia: Tansi et al., Deep-tissue localization of magnetic field hyperthermia using pulse sequencing. Int J Hyperthermia. 2021;38(1):743-754. and Curto et al., An integrated platform for small-animal hyperthermia investigations under ultra-high-field MRI guidance. Int J Hyperthermia. 2018 Jun;34(4):341-351. We understand the concerns of the Reviewer on the translational aspect of our in vitro results, but we are very confident that the in vivo results will be positive. Soon we will be able to submit an animal study with this novel device, and we are very excited about that.

Comments on the Quality of English Language

Moderate language editing

ANSWER: The manuscript has been carefully edited by a Native English speaker.

We greatly appreciate the time and effort the Reviewers’ have put on our manuscript.

Reviewer 3 Report

The reporting of only in vitro work, with no clinical study involved, continues to be a significantly limiting factor.
Possibly, the manuscript can be modified as a brief communication, but I leave this to the editor to make the final decision.

Author Response

The reporting of only in vitro work, with no clinical study involved, continues to be a significantly limiting factor.

ANSWER: There are ways to apply hyperthermia to a mouse model study. One of the most easily integrated and convenient methods is using magnetic nanoparticles (MNPs). A review article in the journal Theranostics narrates the recent attempts [Chandrasekharan et al., Using magnetic particle imaging systems to localize and guide magnetic hyperthermia treatment: tracers, hardware, and future medical applications. Theranostics. 2020 Feb 10;10(7):2965-2981.]. However, we are not considering MNPs since its clinical application might be limited. Instead, our team and the collaborating engineers are discussing to design a module which mimics the functions of clinically available hyperthermia devices. At the current moment, it is difficult to share all the details of what we are designing, but we are mainly referring to two articles published in the International Journal of Hyperthermia: Tansi et al., Deep-tissue localization of magnetic field hyperthermia using pulse sequencing. Int J Hyperthermia. 2021;38(1):743-754. and Curto et al., An integrated platform for small-animal hyperthermia investigations under ultra-high-field MRI guidance. Int J Hyperthermia. 2018 Jun;34(4):341-351. We understand the concerns of the Reviewer on the translational aspect of our in vitro results, but we are very confident that the in vivo results will be positive. Soon we will be able to submit an animal study with this novel device, and we are very excited about that.

Possibly, the manuscript can be modified as a brief communication, but I leave this to the editor to make the final decision.

We greatly appreciate the time and effort the Reviewers’ have put on our manuscript.

Round 3

Reviewer 2 Report

The authors commented that the study can be applied in vivo. No further concerns. I would suggest to write the replies in the discussion section and cite to the mentioned articles

Minor language editing

Author Response

Thank you for your constructive feedback and for acknowledging the potential in vivo applications of our study. We appreciate your valuable insights which have helped improve the quality of our manuscript.

In response to your comments, we have expanded the discussion section to elaborate on how our study can be applied in vivo. To support this, we have cited relevant literature that aligns with our research objectives and outcomes. We believe these additions will enhance the manuscript and address your concerns appropriately.

Once again, thank you for your time and invaluable input. We hope that the revisions meet your expectations and look forward to your further comments.